# Decoupling Adaptation from Modeling with Meta-Optimizers for Meta Learning

## Abstract

Meta-learning methods, most notably Model-Agnostic Meta-Learning (Finn et al., 2017) or MAML, have achieved great success in adapting to new tasks quickly, after having been trained on similar tasks. The mechanism behind their success, however, is poorly understood. We begin this work with an experimental analysis of MAML, finding that deep models are crucial for its success, even given sets of simple tasks where a linear model would suffice on any individual task. Furthermore, on image-recognition tasks, we find that the early layers of MAML-trained models learn task-invariant features, while later layers are used for adaptation, providing further evidence that these models require greater capacity than is strictly necessary for their individual tasks. Following our findings, we propose a method which enables better use of model capacity at inference time by separating the adaptation aspect of meta-learning into parameters that are only used for adaptation but are not part of the forward model. We find that our approach enables more effective meta-learning in smaller models, which are suitably sized for the individual tasks.

## 1 Introduction

Meta-learning or *learning to learn* is an appealing notion due to its potential in addressing important challenges when applying machine learning to real-world problems. In particular, learning from prior tasks but being able to to adapt quickly to new tasks improves learning efficiency, model robustness, etc. A promising set of techiques, Model-Agnostic Meta-Learning (Finn et al., 2017) or MAML, and its variants, have received a lot of interest (Nichol et al., 2018; Lee & Choi, 2018; Grant et al., 2018). However, despite several efforts, understanding of how MAML works, either theoretically or in practice, has been lacking (Finn & Levine, 2018; Fallah et al., 2019).

For a model that meta-learns, its parameters need to encode not only the common knowledge extracted from the tasks it has seen, which form a task-general inductive bias, but also the capability to adapt to new test tasks (similar to those it has seen) with task-specific knowledge. This begs the question: *how are these two sets of capabilities represented in a single model and how do they work together?*

In the case of deep learning models, one natural hypothesis is that while knowledge is represented distributedly in parameters, they can be localized – for instance, lower layers encode task-general inductive bias and the higher layers encode adaptable task-specific inductive bias. This hypothesis is consistent with one of deep learning's advantages in learning representations (or feature extractors) using its bottom layers.

Then we must ask, *in order for a deep learning model to meta-learn, does it need more depth than it needs for solving the target tasks?* In other words, is having a large capacity to encode knowledge that is unnecessary *post*-adaptation the price one has to pay in order to be adaptable? Is there a way to have a smaller (say, less deep) meta-learnable model which still adapts well? This question is of both scientific interest and practical importance – a smaller model has a smaller (memory) footprint, faster inference and consumes less resources.

In this work, through empirical studies on both synthetic datasets and benchmarks used in the literature, we investigate these questions by analyzing how well different learning models can meta-learn and adapt. We choose to focus on MAML due to its popularity.

Our observations suggest depth is indeed necessary for meta-learning, despite the tasks being solvable using a shallower model. Thus, applying MAML to shallower models does not result in successful meta-learning models that can adapt well. Moreover, our studies also show that higher layers are responsible more for adapting to new tasks while the lower layers are responsible for learning task-general features.

Our findings prompt us to propose a new method for meta-learning. The new approach introduces a meta-optimizer which learns to guide the (parameter) optimization process of a small model. The small model is used for solving the tasks while the optimizer bears the burden of extracting the knowledge of how to adapt. Empirical results show that despite using smaller models, the proposed algorithm with small models attains similar performance to larger models which use MAML to meta-learn and adapt.

We note that a recent and concurrent work to ours addresses questions in this line of inquiry (Raghu et al., 2019). They reach similar conclusions through different analysis and likewise, they propose a different approach for improving MAML. We believe our work is complementary to theirs.

## 2 RELATED WORK

Meta-learning, or learning-to-learn, is a vibrant research area with a long and rich history, lying at the intersection of psychology (Maudsley, 1980; Biggs, 1985), neuroscience (Hasselmo & Bower, 1993), and computer science(Schmidhuber, 1987; Thrun & Pratt, 1998) Of particular interest to this manuscript is the line of work concerned with optimization-based meta-learning (OBML) algorithms in the few-shot regime, of which MAML is a particular instance. (Finn & Levine, 2018; Finn et al., 2017; Finn, 2018)

Since its inception, MAML has been widely applied to tackle the few-shot learning challenge, in domains such as computer vision (Lee et al., 2019), natural language processing (Gu et al., 2018), and robotics (Nagabandi et al., 2018). It is also the basis of extensions for continual learning (Finn et al., 2019), single- and multi-agent reinforcement learning (Rothfuss et al., 2018; Al-Shedivat et al., 2017), objective learning (Chebotar et al., 2019), transfer learning (Kirsch et al., 2019), and domain adaptation (Li et al., 2018). Due to its generality, the adaptation procedure MAML introduces – which is the focus of our analysis – has recently been branded as *Generalized Inner Loop Meta-Learning*. (Grefenstette et al., 2019)

While popular in practical applications, relatively few works have analysed the convergence and modelling properties of those algorithms. Finn & Levine (2018) showed that, when combined with deep architectures, OBML is able to approximate arbitrary meta-learning schemes. Fallah et al. (2019) recently provided convergence guarantees for MAML to approximate first-order stationary points for non-convex loss surfaces, under some assumptions on the availability and distribution of the data. Other analyses (empirical or theoretical) have attempted to explain the generalization ability of OBML (Guiroy et al., 2019; Nichol et al., 2018), the bias induced by restricting the number of adaptation steps (Wu et al., 2018), or the effect of higher-order terms in the meta-gradient estimation(Foerster et al., 2018; Rothfuss et al., 2019)

Closely related to our proposed methods are works attempting to improve the adaptation mechanisms of OBML. Meta-SGD (Li et al., 2017) meta-learns per-parameter learning rates, while Alpha MAML (Behl et al., 2019) adapts those learning rates during adaptation via gradient-descent. Meta-Curvature (Park & Oliva, 2019) propose to learn a Kronecker-factored pre-conditioning matrix to compute fast-adaptation updates. Their resulting algorithm is a special case of one of our methods, where the linear transformation is not updated during adaptation. Another way of constructing pre-conditioning matrices is to explicitly decompose all weight matrices of the model in two separate components, as done in T-Nets (Lee & Choi, 2018). The first component is only updated via the evaluation loss, while the second is also updated during fast-adaptation. Warped Gradient Descent (Flennerhag et al., 2019) further extends T-Nets by allowing both components to be non-linear functions. Instead of directly working with gradients, (Chebotar et al., 2019; Xu et al., 2018) suggest to directly learn a loss function which is differentiated during fast-adaptation and results in faster and better learning. Additionally, meta-optimizers have also been used for meta-descent (Sutton, 1981; Jacobs, 1988; Sutton, 1992b;a; Schraudolph, 1999). They can be learned during a pre-training phase

(Andrychowicz et al., 2016; Li & Malik, 2017a;b; Metz et al., 2019b;a;c; Wichrowska et al., 2017) or online (Kearney et al., 2018; Jacobsen et al., 2019; Ravi & Larochelle, 2017).

Our work differentiates from the the above by diagnosing and attempting to address the entanglement between modelling and adaptation in the meta-learning regime. We uncover a failure mode of MAML with linear and smaller models, and propose an effective solution in the form of expressive meta-optimizers.

## 3    ANALYSIS OF MAML

### 3.1    BACKGROUND

In MAML and its many variants (Lee & Choi, 2018; Nichol et al., 2018; Li et al., 2017), we have a model whose parameters are denoted by $\theta$. We would like to optimize $\theta$ such that the resulting model can adapt to new and unseeen tasks fast. To this end, we are given a set of (meta)training tasks, indexed by $\tau$. For each such task, we associate with a loss $\mathcal{L}_\tau(\theta)$.

Distinctively, MAML minimizes the expected task loss after an *adaptation* phase, consisting of a few steps of gradient descent from the model's current parameters. Since we do not have access to the target tasks to which we wish to adapt to, we use the expected loss over the training tasks,

$$\mathcal{L}^{\text{META}} = \mathop{\mathbb{E}}_{\tau \sim p(\tau)} \left[ \mathcal{L}_\tau \left( \theta - \alpha \nabla \mathcal{L}_\tau(\theta) \right) \right] \tag{1}$$

where the expectation is taken with respect to the distribution of the training tasks. $\alpha$ is the learning rate for the adaptation phase. The right-hand-side uses only one step gradient descent such that the aim is to adapt *fast*: in one step, we would like to reduce the loss as much as possible. In practice, a few more steps are often used.

### 3.2    ANALYSIS

**Shallow models can be hard to meta-learn**    Many intuitive explanations for why MAML works exist. One appealing suggestion is that the minimizer of the meta-learning loss $\mathcal{L}^{\text{META}}$ is chosen in such a way that it provides a very good initialization for the adaptation phase; however, if the model is shallow such that $\mathcal{L}_\tau$ is convex in its parameters, then any initialization that is good for fast adapting to one subset of tasks could be bad for another subset of tasks since all the tasks have precisely one global minimizer and those minimizers can be arbitrarily far from each other. When the test tasks are distributed similar to the training tasks, the ideal initialization point has to be the "mean" of the minimizers of the training tasks — the precise definition of the mean is not important, as we will see below.

We illustrate a surprising challenge by studying MAML on a synthetic dataset and the Omniglot task (Lake et al., 2015). Specifically, for the former study, we construct a set of binary classification tasks by first randomly sampling datapoints $w \in \mathbb{R}^{100}$ from a standard Gaussian and use each of them to define a linear decision boundary of a binary classification task. We assume the boundaries pass through the origin and we sample training, validation and testing samples by randomly sampling data points from both sides of the decision boundaries.

By construction, a linear model such as logistic regression is sufficient to achieve very high accuracy on any task. But can MAML learn a logistic regression model from a subset of training tasks that is able to adapt quickly to the test tasks?

Note that due to the random sampling of the training tasks, the average of the minimizers (ie, the samples from the Gaussian distribution) is the origin. Likewise, for a set of test tasks randomly sampled the same way, the origin provides the best initialization by not favoring any particular task.

Figure 1 reports the 1-step post-adaptation accuracy on the test tasks for the meta-learned logistic regression model. Surprisingly, the model fails to perform better than chance. Despite the simplicity of the tasks, logistic regression models are unable to find the origin as an initialization that adapts quickly to a set of test tasks that are similar to the training tasks.

Figure 1 also reports how *being deep* can drastically change the behavior of MAML. There, we add a 4-layer linear network (LinNet) to the logistic regression model (before the sigmoid activation).

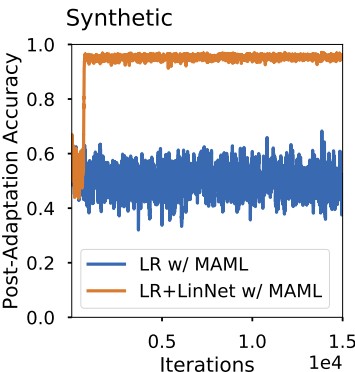 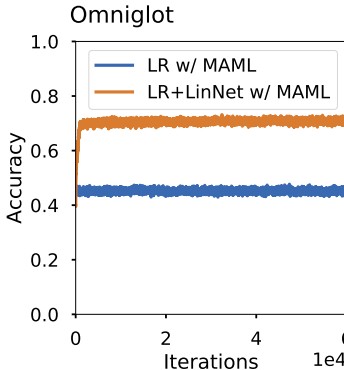

Figure 1: Meta learning results on two datasets: Synthetic binary classification (left) and Omniglot (right). The models to be meta-learned are shallow ones ( logistic regression models for binary classification and multinomial logistic regression) and deep ones – the shallow models with linear networks (LinNet) added to them. While the shallow models do not adapt and achieve chance level accuracies on average, the overparameterized versions attain significantly better results after adaptation.

Note that while the model has the same representational capacity as a linear logistic regression, it is overparameterized and has many local optimizers. As such, MAML can train this model such that its 1-step adaptation accuracy reaches 92% on average.

We observe the same phenomena on meta-learning with MAML on the Omniglot dataset (details of the dataset are given in the Section 5). The shallow logistic regression model achieves 45% accuracy on average (for the 5-way classification tasks) after 2-step adaptation from the meta-learned initialization. However, with a linear network, the adapted model achieves significantly higher accuracy – 70% on average, while having the same modeling capacity as the logistic regression.

In summary, these experiments suggest that even for tasks that are solvable with shallow models, a model needs to have enough depth in order to be meta-learnable and to adapt. We postpone the description of our experiments on nonlinear models to section 5, where we also show that having sufficient depth is crucial for models to be meta-learnable, even when the tasks require fewer layers.

A natural question arises: if *being deep* is so important for meta-learning on even very simple tasks, what different roles, if any, do different layers of a deep network play?

**Depth enables task-general feature learning and fast adaptation** We hypothesize that for deep models meta-learned with MAML, lower layers learn task-invariant features while higher layers are responsible for fast-adaptation.

To examine this claim, we meta-train a model consisting of four convolutional layers (C1 - C4) and a final fully-connected layer (FC) on Omniglot (Lake et al., 2015) and CIFAR-FS (Bertinetto et al., 2019). (Experimental setups are detailed in Section 5.) Once the model has finished meta-training, we perform a layer-wise ablation to study each layer's effect on adaptation. In particular, we iterate over each layer and perform two sets of experiments. In the first, we freeze the weights of the layer such that it does not get updated during fast-adaptation – we call it *freezing only this*. In the second experiment, we freeze all layers but the layer such that this layer is updated during fast-adaptation – we call it *adapting only this*. The left two plots in Figure 2 report the average accuracy over 100 testing tasks from both datasets.

We observe that freezing only the first few lower layers (C1-C3) does not cause noticeable degradation to the post-adaptation accuracy. In fact, as long as the last convolutional layer (C4) is not frozen, post-adaptation accuracy remains unaffected. This indicate that C1-C3 provide information that is task-invariant, while C4 is crucial for adaptation.

This does not mean FC is not important — since adapting C4 requires gradients passing through the FC layer, it cannot be arbitrary. In fact, in the rightmost plot of the figure, C1-C3 are held fixed during adaptation. While C4 and FC are allowed to adapt, and FC is perturbed with noise. When

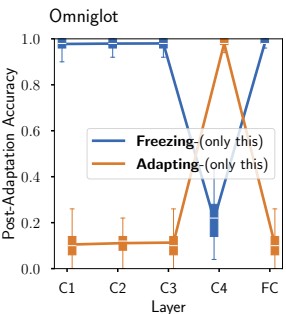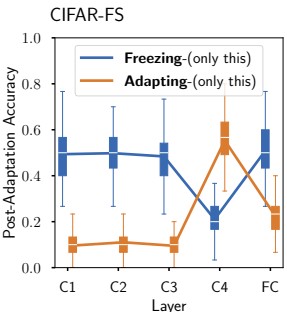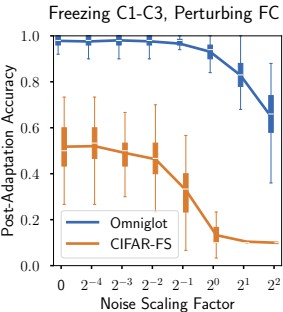

Figure 2: Layer-wise study of each layer's effect on adaptation, in the left and the middle plots. The rightmost plot shows that both C4 and FC have to work together to adapt. See texts for details.

the noise is strong, the performance degrades significantly. Thus, we conclude that both C4 and FC play important roles in the mechanism for fast adaptation. We note that the recent work by Raghu et al. (2019) concurrently reached similar conclusions on the *mini*-ImageNet dataset, using feature similarity-based analyses of the model's representations.

These observations highlight a fundamental issue: the property of being meta-learnable entails more model capacity than being learnable for a specific task. Thus, *MAML can fail on models that lack the capacity to encode both task-general features and adaptation information, even when the models themselves are powerful enough to perform well on each of the tasks used for the meta-learning procedure*. For example, with linear models (e.g. logistic regression), the parameters are forced to overlap and serve both modelling and adaptation purposes. However, as soon as the models are over-parameterized, the extra layers enable meta-learnability. In section 5, we show that this observation also applies to nonlinear models where MAML-trained models quickly lose their performance when the number of layers is reduced.

## 4 META-OPTIMIZER FOR META LEARNING

The previous section has shown that MAML, when applied to deep learning models, meta-learns both task-general features and task-specific adaption parameters at the same time. Since both are represented in a single model where strong dependencies exist between the lower layers and the higher layers, the model needs to be large enough and all its parameters have to be used to solve the test tasks. In some cases, the model has a modelling capacity that is bigger than what the test tasks require. For a model where linear network layers exist, the layers can be collapsed after adaptation so that the actual model used for inference on teh test tasks is small. However, it is not clear how to do so for typical deep models where nonlinearities prevent collapsing the model into smaller ones.

Can we have a different adaptation mechanism such that a smaller model, whose modelling capacity is still adequate for the test tasks, can be adapted to find the minimizers of its loss on the aforementioned tasks?

In this section, we describe a new approach of learning meta-optimizers for meta-learning. The meta-optimizer aims to separate modelling from adaptation. The meta-optimizer transforms the parameter update process for the model so the model can converge fast to its minimizer. How to transform is, however, learned from meta-training tasks. Because of this separation, the model for the task does not have to know how to adapt; it only needs to have enough capacity (roughly, big enough to represent the target task).

We note that classical tools could be used to compress a large model for a given task. However, in the scenario where meta-learning is often used, such tools are unlikely effective. For example, distillation requires a lot of labeled data from the target task, which is not suitable for few-shot learning tasks. Pruning often degrades performance.

### 4.1 LEARNABLE META-OPTIMIZERS

A meta-optimizer – or learnable optimizer – is a parameterized function $U_\xi$ defining the model's parameter updates. For example, a linear meta-optimizer might be defined as:

$$U_\xi(g) = Ag + b, \tag{2}$$

where $\xi = (A, b)$ is the set of parameters of the linear transformation. The objective is to jointly learn the model and optimizer's parameters in the hope of accelerating the minimization of $\mathcal{L}_\tau$. Concretely, we alternate between model and optimizer updates such that their values at adaptation step $t + 1$ are given by:

$$\theta_{t+1} = \theta_t - U_{\xi_t} (\nabla_{\theta_t} \mathcal{L}_\tau (\theta_t)) \tag{3}$$
$$\xi_{t+1} = \xi_t - \alpha \nabla_{\xi_t} \mathcal{L}_\tau (\theta_{t+1}). \tag{4}$$

While this change to the classical optimization pipeline might seem innocuous at first, it bears its own set of computational challenges. First, it is common for modern machine learning models to have millions of parameters. In this case, simply storing the optimizer's parameters becomes infeasible even for the simplest of models, such as the linear one outlined above. Because of this dimensionality issue most of the current literature considers a per-parameter independent update function, thus greatly limiting the expressivity of the meta-optimizer. We propose to address the issue of parameter dimensionality via factorizations of the optimizer's parameters, which we detail in the following subsection.

### 4.2 FACTORIZED META-OPTIMIZERS

To tackle the issue of parameter dimensionality we propose to factorize the optimizer's parameters. To demonstrate this, let us revisit the linear optimizer example above: we assume $g \in \mathbb{R}^k$ to be the vectorization of a matrix gradient $G \in \mathbb{R}^{n \times m}$, such that $\text{vec}(G) = g$ and $m \cdot n = k$. By expressing $A = R^\top \otimes L$ as the Kronecker product of small matrices $R \in \mathbb{R}^{m \times m}$ and $L \in \mathbb{R}^{n \times n}$, the linear update above can be efficiently computed as: (Bernstein, 2018, Section 9.1)

$$U_\xi(g) = \text{vec} (L \cdot G \cdot R) + b, \tag{5}$$

where $b \in \mathbb{R}^k$ and $\xi = (L, R, b)$. In the best case scenario where $m = n = \sqrt{k}$, the above factorization requires $\mathcal{O}(k)$ memory as opposed to $\mathcal{O}(k^2)$ for the non-factored case. Similarly, the matrix-product takes $\mathcal{O}(k\sqrt{k})$ time complexity, while the non-factored version takes $\mathcal{O}(k^2)$.

Note that this linear transformation can be used as a building block to implement more expressive and non-linear meta-optimizers. For example, a fully-connected network meta-optimizer $U_\xi$ is the composition of linear transformations $U_{W_i}$ interleaved with activation function $\sigma$. If the weight matrices $(W_1, \ldots, W_h)$ of the network admit a Kronecker-factorization $W_i = R_i^\top \otimes L_i$, the fully-connected meta-optimizer is imlemented as:

$$U_\xi(g) = U_{W_h} \circ \cdots \circ \sigma \circ U_{W_1} \circ g \tag{6}$$
$$= \text{vec} (L_h \sigma (\ldots \sigma (L_1 \cdot G \cdot R_1) \ldots) R_h), \tag{7}$$

where $\xi = (W_1, \ldots, W_h)$. Such a scheme can be used to implement arbitrary meta-optimizers – such as convolutional or recurrent ones – so long as the architecture involves a composition of linear maps. We refer the reader to Appendix A.3 for pseudo-code and schematics of the model-optimizer loop.

We emphasize that the choice of a Kronecker factorization is arbitrary; many matrix decompositions work equivalently well and result in different modeling and computational trade-offs. For example, using a low-rank Cholesky factorization $A = LL^T$ where $L \in \mathbb{R}^{k \times r}$ allows to interpolate between

Table 1: Accuracies and standard deviation of Meta-Learning of Linear Models

| Dataset | LR w/ MAML | LR + LinNet w/ MAML | LR w/ KLin | LR w/ KFC |
|---|---|---|---|---|
| Synthetic | $51.38\% \pm 3.91$ | $95.77\% \pm 0.41$ | **$99.00\% \pm 0.08$** | **$98.98\% \pm 0.10$** |
| Omniglot | $46.50\% \pm 2.80$ | **$71.31\% \pm 4.52$** | $63.88\% \pm 3.93$ | **$71.56\% \pm 4.40$** |

computational complexity and decomposition rank by tuning the additional hyper-parameter $r$. The Cholesky decomposition might be preferable to the Kronecker one in memory-constrained applications, since $r$ can be used to control the memory requirements of the meta-optimizer. Moreover, such a decomposition imposes symmetry and positiveness on $A$, which might be desirable when approximating the Hessian or its inverse.

In this work, we preferred the Kronecker decomposition over alternatives for three reasons: (1) the computational and memory cost of the Kronecker-product are acceptable, (2) $R^\top \otimes L$ is full-rank whenever $L, R$ are full-rank, and (3) the identity matrix lies in the span of Kronecker-factored matrices. In particular, this last motivation allows meta-optimizers to recover the gradient descent update by letting $R, L$ be the identity.

## 5 EXPERIMENTS

In our experiments, we complement our empirical studies and analysis from Section 3 with further results and address the following question: can we meta-learn smaller (or shallower) models that perform as well as larger (or deeper) ones? To this end, we apply the proposed approach of learning meta-optimizers to the example synthetic dataset, as well as popular benchmark datasets: Omniglot (Lake et al., 2015), *mini*-ImageNet (Ravi & Larochelle, 2017), and CIFAR-FS (Bertinetto et al., 2019). All hyper-parameters and additional experimental details are available in Appendix A.1.

Unless otherwise noted, all models are trained using MAML. For models that use learnable deep optimizers, we meta-learn both model and optimizer parameters. Similarly, both sets of parameters are adapted during fast-adaptation.

To differentiate, we add the name of the optimizer to the model. For example, LR w/ KLin corresponds to a logisitic regression model with the Kronecker-factorized linear transformation from the previous section. Similarly, w/ KFC indicates a model optimized by **K**ronecker-factored **F**ully-**C**onnected network in the previous section.

### 5.1 META-LEARNING LINEAR MODELS

We adopt the same setting as in section 3 where we study the synthetic data and the Omniglot dataset. For the Omniglot dataset, we consider the 1-shot, 5-way setup of Finn et al. (2017) and let the model adapt for 2 steps. Recall that the logistic regression model does not meta-learn while overparameterized logistic regression models (ie, models with linear networks) do.

Table 1 reports the final post-adaptation testing accuracies and their standard deviations. As can be observed, while logistic regression (LR) cannot be meta-learned, overparameterized models (LinNet) as well as LR with our meta-optimizers (LR w/ KLin and LR w/ KFC) can all meta-learn while our approaches lead to similar or better results compared to LinNet.

Note that since linear networks can always be collapsed into a single weight matrix and bias term, any increase in performance is the consequence of better adaptation, rather than better modelling.

### 5.2 NON-LINEAR MODELS

We examine whether our meta-optimizers can meta-learn successfully with smaller but deep nonlinear models to match bigger deep models. We focus on two datasets: the Omniglot where the setting is 10-way classification with 5-shots and 4 adaptation steps, using the original 4-layer convolutional network (CNN) of Finn et al. (2017), and the CIFAR-FS dataset (Bertinetto et al., 2019), doing 10-way classification with 3-shots and 2 adaptation steps. The model is a CNN similar to the one for the Omniglot, but only uses 32 hidden units as opposed to 64.

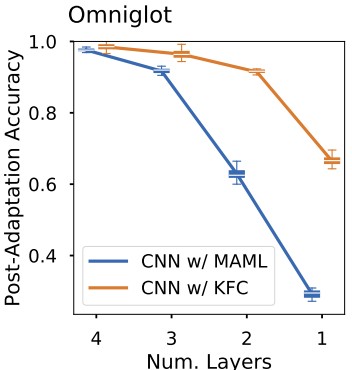 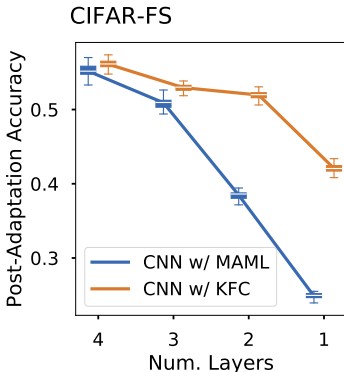

Figure 3: The effect of the number of convolutional layers on adaptation performance. While the model becomes smaller, the performance drops. However, our approach degrades in a slower pace, as the adaptation is shared by both the model and the meta-optimizer, thus leaving more freedom for the model to represent the tasks.

Table 2: Post-adaptation accuracies by different meta-optimizers adapting a small CNN.

| Task | UPPER BOUND | Meta-Learner Architecture | | | |
| | | W/ MAML | W/ MSGD | W/ MC | W/ KFC |
|---|---|---|---|---|---|
| Omniglot | $95.09\% \pm 2.32$ | $60.94\% \pm 1.54$ | $69.25\% \pm 0.81$ | $85.70\% \pm 0.91$ | $\mathbf{91.13\% \pm 0.93}$ |
| CIFAR-FS | $63.94\% \pm 3.89$ | $38.44\% \pm 0.87$ | $45.96\% \pm 1.73$ | $\mathbf{52.89\% \pm 1.43}$ | $\mathbf{52.94\% \pm 1.33}$ |
| *mini*-ImageNet | $58.45\% \pm 5.58$ | $18.55\% \pm 0.46$ | $26.59\% \pm 1.43$ | $\mathbf{29.74\% \pm 0.74}$ | $27.40\% \pm 0.60$ |

We report numerical results in detail in Table A2 in the Appendix, which are graphically presented in Figure 3. We vary the number of convolutional layers. As the number of the layers decrease, the adaptation performance by both MAML and our approach (KFC) decrease. However, our approach has a much slower degradation rate. In fact, our approach is generally able to adapt a smaller model to the same level of performance as a MAML-trained model with an additional layer.

### 5.3 COMPARISON TO OTHER META-OPTIMIZERS

We also compare our approach of learning meta-optimizers to other approaches, notably Meta-SGD (Li et al., 2017) and Meta-Curvature (Park & Oliva, 2019). Both of these methods attempt to approximate the expected task-loss landscape curvature and can be seen as ablations of our methods; Meta-Curvature corresponds to `KLin` without adapting the optimizer, while Meta-SGD approximates Meta-Curvature with a diagonal matrix.

To ease the computational burden, we use a smaller CNN with 2 convolutional layers (SCNN) model for adaptation. As an upper bound on adaptation performance, we train the model on individual tasks to convergence and report the average final accuracy. Additionally, we include results on the *mini*-ImageNet dataset for all methods, in the 10-way 1-shot with 5 adaptation steps setting (Ravi & Larochelle, 2017).

Table 2 reports post-adaptation testing accuracies. All methods of learning optimizers are able to adapt better than SCNN w/ MAML. In particular, our approachs perform the best and comes closest to approaching the upper bound of performance (SCNN).

## 6 CONCLUSION

We introduce our approach by analyzing the success and failure modes of optimization-based meta-learning methods. Namely, we find that, when successful, these methods tend to learn task-general features in early layers and adaptable parameters/update functions in the later layers. Moreover, we find that this learning fails when model size is reduced, indicating that optimization-based meta-learning methods rely on the ability to encode task-general features and/or adaptable parameters,

even when the model itself is adequate for learning on the individual tasks. As such, we introduce our method for decomposing modelling from adaptation using factored meta-optimizers. These meta-optimizers enable the forward model to use more capacity on learning task-specific features, while the expressiveness of their updates allows the forward model to adapt quickly to different tasks. We find that our approach is able to enable successful meta-learning in models that do not work with traditional optimization-based meta-learning methods.

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

## A APPENDIX

### A.1 EXPERIMENTAL DETAILS AND HYPER-PARAMETERS

The following paragraphs describe the data-generation process, model architectures, and learning hyper-parameters for experiments in Section 5. Meta-optimizers's parameter matrices are always initialized to the identity.

#### A.1.1 LINEAR MODEL EXPERIMENTS

**Synthetic Binary Classification** The task decision boundaries $w \sim \mathcal{N}(0.0, I_{100})$ are sampled from standard Gaussian. Each one of the 1,000 data point $x \sim \mathcal{N}(0.0, I_{100})$ is also sampled from the standard Gaussian, and is assigned a label from $\{-1, 1\}$ based on the sign of $w^\top x$. All models are trained for 30,000 iterations of stochastic gradient descent (without momentum) so as to minimize the 1-step MAML loss $\mathcal{L}^{\text{MAML}}$. The task loss $\mathcal{L}(\theta)$ is the binary cross-entropy. Each stochastic gradient is estimated on a single task. (i.e. the meta-batch size is 1.)

For the logistic regression (LR) results, we used meta and adapatation learning rates of 0.01 and 0.5. The linear network (LR+LinNet) consists of 3 hidden layers of 64 units, and is trained with both meta- and fast-adaptation learning rates set to 1.9. Experiments using the Kronecker-factored linear meta-optimizer also use the same meta- and fast-adaptation learning rate, set to 3.33. The meta-optimizer itself consists of 10,001 parameters. ($m = 100, n = 1$) The Kronecker-factored fully-connected meta-optimizer uses a single hidden layer with rectified linear-units (ReLU) activation functions, for a total of 20,002 parameters. ($m = 100, n = 1$) It has both learning rates set to 3.9. None of the meta-optimizers use a bias term.

**Omniglot** Our Omniglot experiments exactly replicate the setup of Finn et al. (2017). Of the 1623 classes, we designate 1,200 for meta-training, 100 for validation, and 423 for testing. We then generate 20,000 training tasks, 600 validation/testing tasks, each consisting of five classes containing a single character image, possibly rotated by 90, 180, or 270 degrees. All models are trained using Adam for 60,000 iterations, an iteration consisting of 32 tasks for which the model is allowed 2 steps of fast-adaptation.

The logistic regression model uses a meta-learning rate of 0.0005 and an adaptation learning-rate set to 1.0. The linear network uses 4 hidden layers of 256, 128, 64, and 64 units and flattens the character images into a 784-dimensional vector before processing them. The meta-learning rate was set to 0.0005 and the adaptation learning rate to 0.08. For the KLin experiment (LR w/ KLin), we use a meta-learing rate of 0.003 and set the adaptation learning rate to 0.9. The KLin optimizer consists of 614k parameters, with $m = 784, n = 5$. The KFC meta-optimizer (LR w/ KFC) consists of 4 hidden layers (2.5M parameters) with ReLU activations, whose meta and adaptation learning rates are set to 0.001 and 0.01, respectively. Again, none of the meta-optimizer use biases.

#### A.1.2 NONLINEAR MODEL EXPERIMENTS

For the non-linear model experiments, we learn one KFC optimizer per layer of the model. Each KFC optimizer is based on the same architecture, but learns its own set of parameters. Moreover, we separate processing of the magnitude and direction of the model's gradient as follows: the gradient is initially scaled so as to have unit norm before being fed to the KFC network. Once the normalized gradient has been processed, it is rescaled by the initial normalizing constant times a learnable factor. (This learnable factor is initialized to 1.) We found this architecture to help in situations where the model is allowed several adaptation steps. For convolutional models, we flatten the height and width weight dimensions such that the optimizer is shared across filters of a same layer. (i.e. if the convolutional layer has shape NxCxHxW, we have $m = C, n = HW$.) Note that, again, none of the meta-optimizers use a bias term.

**Omniglot** The dataset replicates the processing of linear model experiments, detailed above. Instead of 5 ways and 1 shot, we use 10 ways and 5 shots. All models are allowed 4 steps of fast adaptation.

The 4-layer CNN network (CNN w/ MAML) uses a meta-learning rate of 0.003, and an adaptation learning rate of 0.5. Its architecture replicates the one described in Finn et al. (2017). The 2-layer CNN (SCNN w/ MAML) uses the same first two layers and same input dimensionality to the FC layer as the 4-layer CNN, with meta and adapation learning rates set to 0.003 and 0.8, respectively. The 4-layer CNN has a total of 112,586 parameters, while the 2-layer only 38,474. We used the same learning rates for the Meta-SGD, Meta-Curvature, and KFC optimizers: a meta learning rate of 0.003, and an adaptation learning rate of 0.5. The KFC architecture consists of 4 layers, such that the meta-optimizers contains a total of 134,171 parameters for the 2-layer CNN model and 267,451 parameters for the 4-layer CNN.

**CIFAR-FS**    We obtained the splits created by Bertinetto et al. (2019) and exactly reproduced their preprocessing setting for our experiments on CIFAR-FS. We also split the 100 classes in 64 training, 16 validation, and 20 testing classes and generate 20,000 training tasks, and 600 validation and testing tasks. We consider the 10-ways 3-shots setup with 2 fast-adaptation steps. As opposed to Bertinetto et al. (2019), our model closely resembles the one of our Omniglot experiments. Its main difference with the model described in prior work is the absence of max-pooling layers, and averaging of the last two convolutional feature dimensions before the fully-connected layer. Doing so allows us to conduct our ablation studies on the effect of our meta-optimizers with different number of convolutional layers, while keeping the input dimensionality of the fully-connected layer constant.

The 4- and 2-layer CNNs use meta- and fast-adaptation learning rates of 0.003 and 0.7, respectively. The 4-layer CNN has a total of 29,226 parameters, while the 2-layer only 10,602. Meta-SGD, Meta-Curvature, and KFC also use a 0.003 meta-learning rate, but the adaptation learning rate is decreased to 0.5. The KFC optimizer consists 4 layers with ReLU activations, for a total of 69,325 parameters for the 4-layer CNN model and 35,117 parameters for the 2-layer CNN.

***mini*-ImageNet**    As for the Omniglot experiment, we replicate the *mini*-ImageNet setting and model from Finn et al. (2017). We consider the 10-ways, 1-shot and 5 fast-adaptation steps setting.

The 4-layer CNN and 2-layer CNN both use a meta-learning rate of 0.0005 and a fast-adaptation learning rate of 0.07. The 4-layer CNN has a total of 36,906 parameters, while the 2-layer only 18,282. The Meta-SGD, Meta-Curvature, and KFC meta-optimizer use the same meta-learning rate, but a smaller fast-adaptation learning rate set to 0.1 for Meta-SGD, 0.09 for Meta-Curvature, and 0.005 for KFC. For the 2-layer CNN model, the 4-layer KFC optimizer consists of approximately 2.60M parameters, while for the 4-layer CNN model it consists of 2.63M parameters.

## A.2 ADDITIONAL EXPERIMENTAL RESULTS

This section provides additional experimental evidence supporting the claims in the main text.

### A.2.1 EFFECT OF WIDTH ON ADAPTABILITY

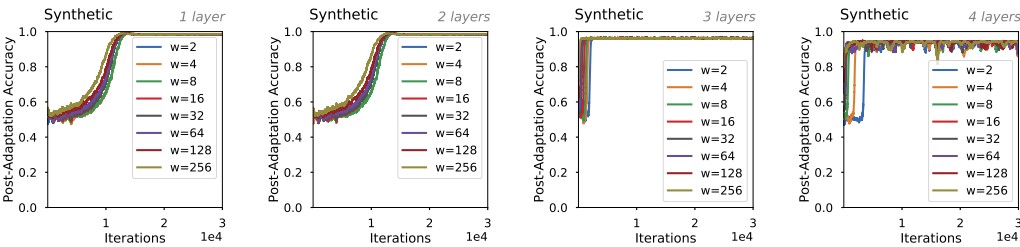

Figure A.1: Effect of width on adaptability on the synthetic binary classification setting. We vary the width and depth of linear networks and report their post-adaptation accuracy. As long as the model has one or more hidden layers, the model is able to adapt to the tasks regardless of the layers' width.

In Section 3, we argue that depth is a curcial ingredient to enable gradient-based meta-learning. In particular, we show that even on simple, linearly separable binary classification tasks a logistic regression model (i.e. with no hidden layer) fails to solve the meta-learning problem. Using the same toy experiment, we now argue that width plays a relatively less important role with respect to adaptability.

In Figure A.2, we plot the post-adaptation accuracy for different number of hidden layers, and vary their width $w = 2, 4, \ldots, 256$. As can be seen, all models are able to eventually solve the meta-learning problem regardless of width, so long as the model has one or more hidden layers.

### A.2.2 DO LAYERS C1-C3 BEHAVE SIMILARLY TO FC ?

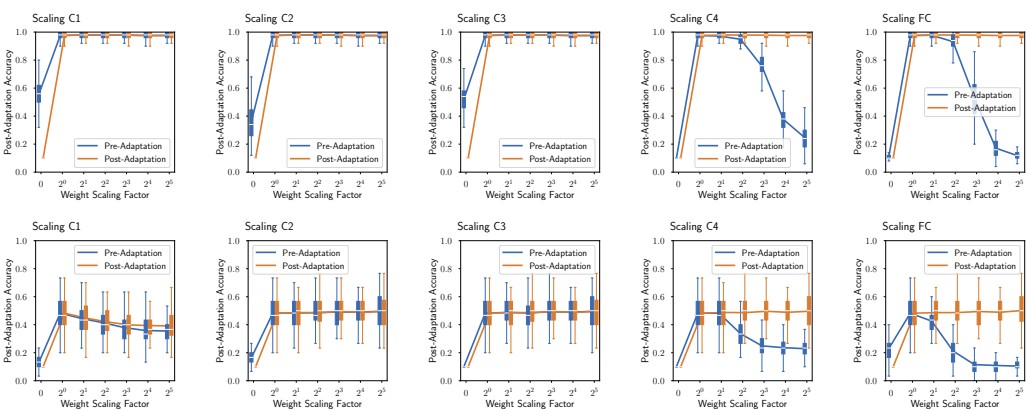

Figure A.2: Weight scaling experiment for the meta-trained models on Omniglot (top) and CIFAR-FS (bottom). We observe that the modelling layers (C1-C3) are insensitive to parameter scaling regardless of whether it is applied pre- or post-adaptation, indicating they serve modelling purposes. On the other hand, parameter scaling on the adaptation layers (C4 & FC) quickly degrades post-adaptation accuracy if applied pre-adaptation, but has virtually no effect if applied post-adaptation. In turn, this underlines the importance of these layers for adaptation.

In Section 3, we claim that early layers in deep meta-trained models are purposed for modelling, while latter layers are responsible for fast-adaptation. This conclusion is reached through the ar-

gument that (a) layers C1-C3 can be completely omitted from the adaptation dynamics, and (b) perturbing the last FC layer results in large post-adaptation accuracy degradation.

We now present further evidence that both layer groups (i.e. C1-C3 and C4-FC) indeed serve different purposes. For each layer of a meta-trained model, we multiply the weights of the selected layer by a constant factor either pre- or post-adaptation. Intuitively, layers that affect the fast-adaptation update will incur a large post-adaptation accuracy penalty as the scaling magnitude increases, due to the value of their weights playing an important part in the computation of the adaptation gradient. On the other hand, a constant scaling of the weights should have little effect on the modelling ability of the network, due to the relative magnitude of activation being maintained. (Note that this intuition is somewhat blurred for our models, as they make use of batch normalization.)

Figure A.2 reports post-adaptation accuracy for those experiments. As predicted, scaling the weights of C1-C3 has little impact on accuracy, whether applied pre- or post-adaptation. Clearly, those layers do not affect the computation of the fast adaptation update. Similarly, scaling the weights of C4-FC post-adaptation does not impact the modelling ability of the network, and post-adaptation accuracy does not suffer. However, the same rescalings prove to be disastrous when applied pre-adaptation; the model is unable to adapt and accuracy drops by as much as 85% for Omniglot and 40% for CIFAR-FS. Evidently, those layers are central to successful fast-adaptation.

### A.2.3 Additional Methods for the Non-linear Model Experiments

In this section, we detail additional methods and apply them to the non-linear models of Section 5.

**Cholesky-factored Meta-Optimizers** As alluded in Section 4, the choice of Kronecker decomposition is arbitrary. We now describe an alternative implementation of meta-optimizers, based on a low-rank Chholesky decomposition. The rank $r$ Cholesky decomposition consists of a matrix $L \in \mathbb{R}^{n \times r}$, such that a rank $r$ symmetric positive definite matrix $A \in \mathbb{R}^{n \times n}$ is expressed as $A = LL^\top$. Using this decomposition, we can write the linear transformation of Section 4 as:

$$Ag + b = (LL^\top)g + b = L(L^\top g) + b, \tag{8}$$

where $b \in \mathbb{R}^{n \times 1}$ is the bias term and $g \in \mathbb{R}^{n \times 1}$ is the vectorization of a weight's gradient. As for the Kronecker product, the latter expression of the linear transformation does not require to construct a large $n \times n$ matrix. Additionally, multiple linear transformations can be chained or interleaved by activation functions in order to obtain more expressive and adaptable meta-optimizers.

A major attribute of the Cholesky decomposition is its flexibility in terms of memory requirements. For example, by letting $r = 1$ we force the meta-optimizer to have as many weights as the model. This attribute can in turn be a disadvantage: as we show in our experiments below, $r$ plays an important role in the expressivity of the meta-optimizer, and should be treated as an additional hyper-parameter to tune. A second inconvenience stems from the choice of initialization scheme of the meta-optimizer; for the Kronecker product, we initialize the Kronecker factors to the identity, thus recovering hypergradient descent in the first meta-training batch. However, since the identity matrix $I_n$ does not lie in the span of low-rank Cholesky factorizable matrices, the initialization scheme becomes another decision left to the practitioner. For our subsequent experiments, we find that letting $L$ follow a Glorot initialization (i.e. $L_{ij} \sim \mathcal{N}(0, \text{gain} \cdot \sqrt{\frac{1}{n}})$) to work well.

Table A1: Post-adaptation accuracies by different meta-optimizers adapting a small CNN.

| Task | Meta-Learner Architecture | | | | |
|---|---|---|---|---|---|
| | w/ MAML | w/ PRUNE | w/ CFC1 | w/ CFC10 | w/ KFC |
| Omniglot | $60.94\% \pm 1.54$ | $70.06\% \pm 8.87$ | $72.62\% \pm 1.12$ | $79.56\% \pm 1.24$ | $\mathbf{91.13\% \pm 0.93}$ |
| CIFAR-FS | $38.44\% \pm 0.87$ | $10.00\% \pm 0.05$ | $32.11\% \pm 0.98$ | $47.81\% \pm 0.91$ | $\mathbf{52.94\% \pm 1.33}$ |
| *mini*-ImageNet | $18.55\% \pm 0.46$ | $12.20\% \pm 0.13$ | $16.13\% \pm 1.29$ | $21.48\% \pm 1.57$ | $\mathbf{27.40\% \pm 0.60}$ |

We compare the low-rank Cholesky-factored meta-optimizers to the Kronecker ones in Table A1. We denote by CFC1 the rank-1 Cholesky-factored fully-connected network, while CFC10 indicates

the rank-10 version. CFC10 contains approximately as many parameters as KFC. (c.f. Table **??**) While models trained with CFC do not outperform their KFC counter-parts, they offer a competitive alternative to existing methods.

**Effect of Pruning**    We study the effect of post-adaptation pruning of a meta-trained model. Ideally, pruning provides an attractive alternative to explicitly separating adaptation from modelling for learning lean models: practitioners could meta-train a larger model, adapt it to the desired target task, and finally prune it. We implement this pipeline by pruning weights smaller than a predefined threshold and report results in Figure A.3. As can be observed, pruning the 4-layer CNN models to obtain the equivalent number of parameters to a 2-layer SCNN can be catastrophic. On Omniglot, the pruned model reaches 70.06% post-adaptation accuracy, more than 20% lower than using meta-optimizers. On CIFAR-FS and *mini*-ImageNet, pruning can barely outperform chance prediction, reaching on average 10.00% and 12.20%, respectively. These results are also available in Table A1, in the w/ PRUNE column.

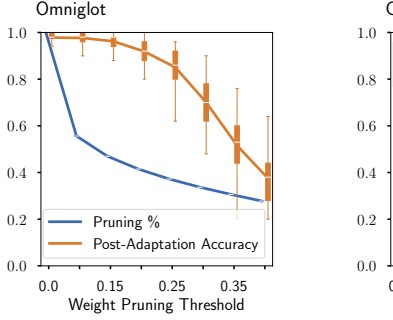 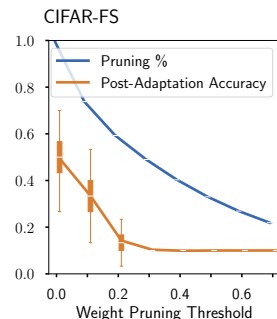 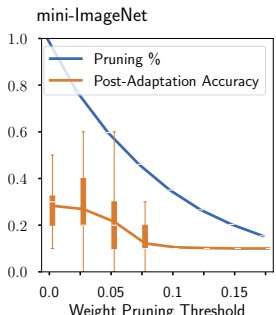

Figure A.3: Effect of pruning weights smaller than a given threshold on the 4-layer networks for Omniglot (left), CIFAR-FS (middle), and *mini*-ImageNet (right). The respective pruning thresholds to obtain the same number of parameters as the SCNN are 0.30 (Omniglot), 0.5 (CIFAR-FS), and 0.075 (*mini*-ImageNet). For those pruning values, the accuracies are much lower than the ones obtained by separating the adaptation mechanism from the model.

## A.3 ADDITIONAL VISUALIZATIONS

In this section, we provide visualizations of experimental results complementary to the ones presented in the main text.

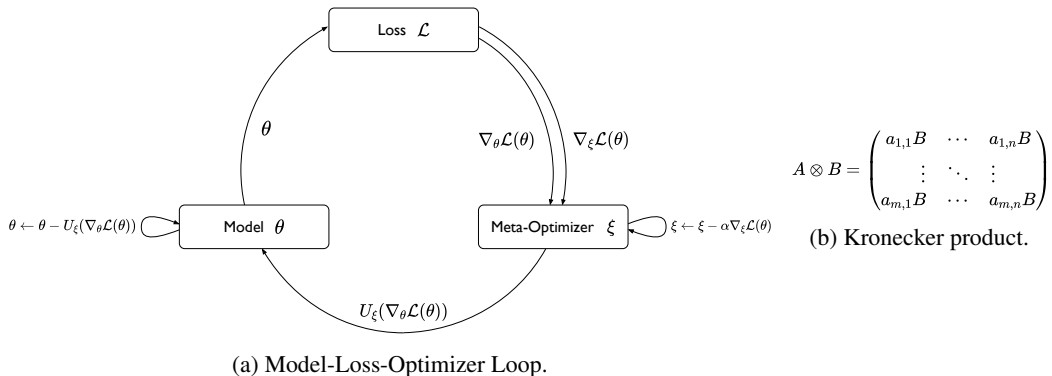

(a) Model-Loss-Optimizer Loop.

(b) Kronecker product.

Figure A.4: Schematic of the model-loss-optimizer loop and of the Kronecker product.

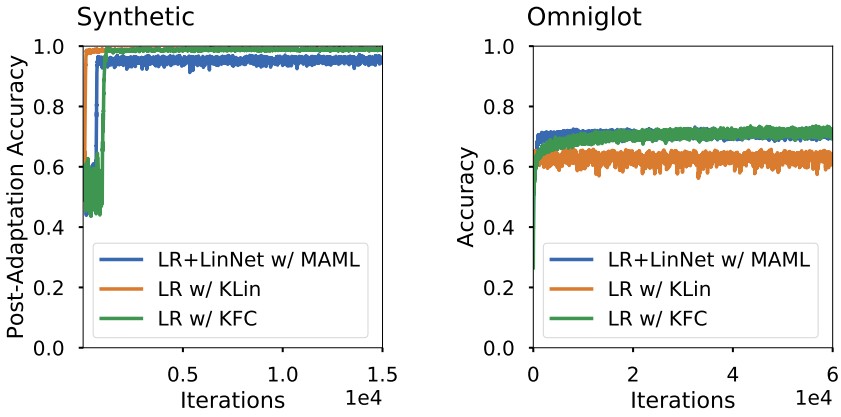

Figure A.5: Evolution of test accuracies along training for our methods with linear models on the synthetic data binary classification (left) and Omniglot (right).

Table A2: Accuracies and standard deviation of Meta-Learning of Non-Linear Models

| | Omniglot | | CIFAR-FS | |
|---|---|---|---|---|
| Num. Layers | CNN | CNN w/ KFC | CNN | CNN w/ KFC |
| 1 | $29.20\% \pm 1.13$ | $\mathbf{66.45\% \pm 1.42}$ | $24.86\% \pm 0.47$ | $\mathbf{42.02\% \pm 0.52}$ |
| 2 | $60.94\% \pm 1.54$ | $\mathbf{91.13\% \pm 0.93}$ | $38.44\% \pm 0.87$ | $\mathbf{52.94\% \pm 1.33}$ |
| 3 | $91.69\% \pm 0.70$ | $\mathbf{96.57\% \pm 0.86}$ | $50.91\% \pm 0.80$ | $\mathbf{53.07\% \pm 0.58}$ |
| 4 | $97.56\% \pm 0.45$ | $\mathbf{98.49\% \pm 0.41}$ | $55.03\% \pm 1.23$ | $\mathbf{55.97\% \pm 0.59}$ |

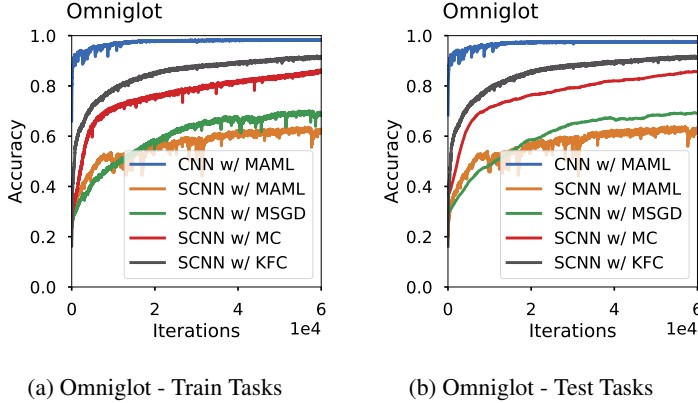

(a) Omniglot - Train Tasks

(b) Omniglot - Test Tasks

Figure A.6: Evolution for the training (left) and testing (right) accuracies of the non-linear models on Omniglot.

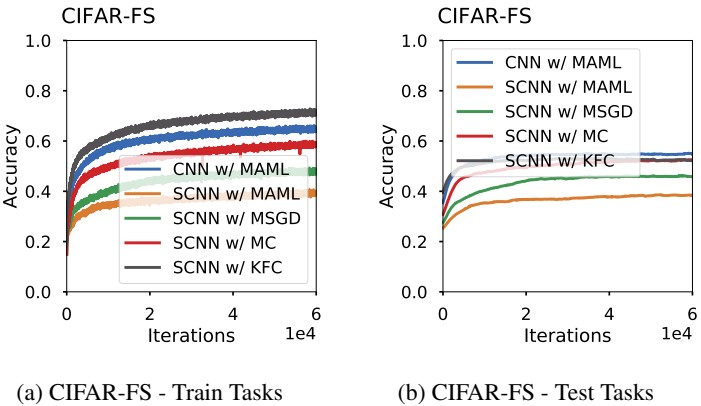

(a) CIFAR-FS - Train Tasks

(b) CIFAR-FS - Test Tasks

Figure A.7: Evolution for the training (left) and testing (right) accuracies of the non-linear models on CIFAR-FS.

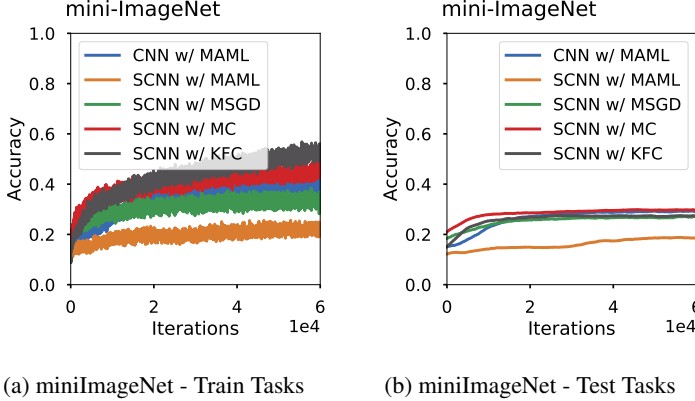

(a) miniImageNet - Train Tasks

(b) miniImageNet - Test Tasks

Figure A.8: Evolution for the training (left) and testing (right) accuracies of the non-linear models on *mini*-ImageNet .

---

**Algorithm 1** Meta-Learning with Meta-Optimizers

---

**Require:** Fast learning rate $\alpha$, Initial parameters $\theta^{(\text{init})}$ and $\xi^{(\text{init})}$, Optimizer `Opt`
 1: **while** $\theta^{(\text{init})}, \xi^{(\text{init})}$ not converged **do**
 2:    Sample task $\tau \sim p(\tau)$
 3:    $\theta_1 = \theta^{(\text{init})}, \quad \xi_1 = \xi^{(\text{init})}$
 4:    **for** step $t = 1, \ldots, T$ **do**
 5:       Compute loss $\mathcal{L}_\tau(\theta_t)$
 6:       Compute gradients $\nabla_{\theta_t} \mathcal{L}_\tau(\theta_t)$ and $\nabla_{\xi_t} \mathcal{L}_\tau(\theta_t)$
 7:       Update the meta-optimizer parameters $\xi_{t+1} = \xi_t - \alpha \nabla_{\xi_t} \mathcal{L}_\tau(\theta_t)$
 8:       Compute the model update $U_{\xi_{t+1}}(\nabla_{\theta_t} \mathcal{L}_\tau(\theta_t))$
 9:       Update the model parameters $\theta_{t+1} = \theta_t - U_{\xi_{t+1}}(\nabla_{\theta_t} \mathcal{L}_\tau(\theta_t))$
10:    **end for**
11:    Update model and meta-optimizer initializations
12:       $\theta^{(\text{init})} \leftarrow \theta^{(\text{init})} - \texttt{Opt}(\nabla_{\theta^{(\text{init})}} \mathcal{L}_\tau(\theta_T))$
13:       $\xi^{(\text{init})} \leftarrow \xi^{(\text{init})} - \texttt{Opt}(\nabla_{\xi^{(\text{init})}} \mathcal{L}_\tau(\theta_T))$
14: **end while**

---

## A.4 COMPUTATIONAL METRICS

This section provides computational metrics for the non-linear experiments in Table A3. All metrics are measured using a single NVIDIA Titan XP.

Table A3: Computational metrics for each of the methods. The timing metrics measure the time to compute one meta-gradient step on an entire meta-batch. Inference time is identical across methods for models of the same size.

| Task | Method | Model Parameters | Optimizer Parameters | Time | Accuracy |
|------|--------|------------------|----------------------|------|----------|
| Omniglot | CNN | | | | |
| | W/ MAML | 112,586 | 0 | 1.09s | 97.56% |
| | W/ KFC | 112,586 | 267,451 | 10.35s | **98.49%** |
| | SCNN | | | | |
| | W/ MAML | 38,474 | 0 | 0.58s | 60.94% |
| | W/ KFC | 38,474 | 134,171 | 5.44s | 91.13% |
| | W/ CFC1 | 38,474 | 8,905 | 4.30s | 72.62% |
| | W/ CFC10 | 38,474 | 66,379 | 4.36s | 79.56% |
| CIFAR-FS | CNN | | | | |
| | W/ MAML | 29,226 | 0 | 0.63s | 55.03% |
| | W/ KFC | 29,226 | 69,325 | 2.04s | **55.97%** |
| | SCNN | | | | |
| | W/ MAML | 10,602 | 0 | 0.25s | 38.44% |
| | W/ KFC | 10,602 | 35,117 | 1.08s | 52.94% |
| | W/ CFC1 | 10,602 | 4,185 | 0.83s | 32.11% |
| | W/ CFC10 | 10,602 | 34,317 | 0.84s | 47.81% |
| *mini*-ImageNet | CNN | | | | |
| | W/ MAML | 36,906 | 0 | 0.25s | 29.20% |
| | W/ KFC | 36,906 | 2.63M | 1.75s | **30.81%** |
| | SCNN | | | | |
| | W/ MAML | 18,282 | 0 | 0.14s | 18.55% |
| | W/ KFC | 18,282 | 2.60M | 0.94s | 27.40% |
| | W/ CFC1 | 18,282 | 42,585 | 0.69s | 16.13% |
| | W/ CFC10 | 18,282 | 349,197 | 0.71s | 21.48% |

