# OpenReview forum: "Decoupling Adaptation from Modeling with Meta-Optimizers for Meta Learning"
_ICLR.cc/2020/Conference — Reject_

### Official Review · AnonReviewer1 · 2019-10-23
**Official Blind Review #1**

**Rating:** 3

**Review:**

This paper investigated the effect of depth on the meta-learning model.
The paper mainly studies through experimental means and does not have mathematical analysis to demonstrate. In this way of analysis, a large number of experiments are necessary. In addition to ensuring a large number of experiments, it is necessary to ensure the diversity of methods. This article only studied MAML, therefore, the conclusion of the experimental inquiry cannot convince me.
For the experimental part, I am afraid the results are also weak. For example, please notice that many meta-learning models have proposed. I believe authors should compare more existing works to demonstrate the superiority of the proposed one.

[Update after rebuttal period]
It may seem reasonable that depth enables task-general feature learning. However, in fact, it is not true. The major reason for people to think that the receptive field becomes very large after multiple pooling operation. This is true but not the reason for good performance in feature learning. Because of back-propagation, the feature extraction layers can be trained well to extract features from objects of different scales. The major reason for poor performance in feature learning is that the header that creates an object template is not well trained for objects of different scales. As a result, I still keep the confusion in terms of the effectiveness of the proposed method.



**Experience Assessment:**

I have published in this field for several years.

**Review Assessment: Checking Correctness Of Derivations And Theory:**

N/A

**Review Assessment: Checking Correctness Of Experiments:**

I carefully checked the experiments.

**Review Assessment: Thoroughness In Paper Reading:**

I read the paper at least twice and used my best judgement in assessing the paper.

---

> ### Author Response · Authors · 2019-11-15
> **Response to AnonReviewer1**
>
> This paper studies MAML, as it is a “seminar and widely followed work” (Reviewer#2). It has been extended [1-3] and widely applied across multiple subfields. (e.g. computer vision [4], robotics [5], and dialogue systems [6]). Particularly relevant to the reviewer’s concern, there have been multiple empirical and theoretical works dedicated solely to the study of MAML [7-11]. Yet, the understanding of why and how MAML works is far from being complete. Thus, the paper sets to make progress in this direction, hypothesizing “depth” as an (unexplored) and important aspect to MAML.
>
> Note that other approaches [12-18] are either too different to analyse using the proposed empirical approaches or simply do not fit the few-shot meta-learning paradigm. Note that when possible, we do compare against methods having a similar flavour as the one we propose. (i.e. MetaSGD, MetaCurvature). However, to make our efforts more precise, we are happy to make it clear that this work specifically addresses MAML (and its alike).
>
> Regarding the depth/breadth of our experiments, our paper currently features results on 1 synthetic and 3 popular computer vision datasets, which is as much or more than similar submitted/published works [9, 16, 18]. Maybe more importantly, the results for all experimental settings agree with each other, and some (e.g. the freezing experiments) were independently discovered by other researchers. Altogether, we believe this is a testament to their generality and replicability.
>
> We hope that in light of the above, the content of our paper has become more appealing; our goal is not to propose a new state-of-the-art method, but rather to shine some light on the underlying dynamics of a popular meta-learning algorithm.

---

> > ### Author Response · Authors · 2019-11-15
> > **References**
> >
> > Bibliography:
> > 1. Finn C, Rajeswaran A, Kakade S, Levine S. "Online Meta-Learning". 22 Feb 2019. http://arxiv.org/abs/1902.08438
> > 2. Rajeswaran A, Finn C, Kakade S, Levine S. "Meta-Learning with Implicit Gradients". 10 Sep 2019. http://arxiv.org/abs/1909.04630
> > 3. Triantafillou, Eleni, Tyler Zhu, Vincent Dumoulin, Pascal Lamblin, Kelvin Xu, Ross Goroshin, Carles Gelada, Kevin Swersky, Pierre-Antoine Manzagol, and Hugo Larochelle. 2019. “Meta-Dataset: A Dataset of Datasets for Learning to Learn from Few Examples.” arXiv [cs.LG]. arXiv. http://arxiv.org/abs/1903.03096.
> > 4. Lee, Kwonjoon, Subhransu Maji, Avinash Ravichandran, and Stefano Soatto. 2019. “Meta-Learning with Differentiable Convex Optimization.” arXiv [cs.CV]. arXiv. http://arxiv.org/abs/1904.03758.Lee, Kwonjoon, Subhransu Maji, Avinash Ravichandran, and Stefano Soatto. 2019. “Meta-Learning with Differentiable Convex Optimization.” arXiv [cs.CV]. arXiv. http://arxiv.org/abs/1904.03758.
> > 5. Nagabandi, Anusha, Ignasi Clavera, Simin Liu, Ronald S. Fearing, Pieter Abbeel, Sergey Levine, and Chelsea Finn. 2018. “Learning to Adapt in Dynamic, Real-World Environments Through Meta-Reinforcement Learning.” arXiv [cs.LG]. arXiv. http://arxiv.org/abs/1803.11347.
> > 6. Mi, Fei, Minlie Huang, Jiyong Zhang, and Boi Faltings. 2019. “Meta-Learning for Low-Resource Natural Language Generation in Task-Oriented Dialogue Systems.” arXiv [cs.CL]. arXiv. http://arxiv.org/abs/1905.05644.
> > 7. Finn, Chelsea, and Sergey Levine. 2017. “Meta-Learning and Universality: Deep Representations and Gradient Descent Can Approximate Any Learning Algorithm.” arXiv [cs.LG]. arXiv. http://arxiv.org/abs/1710.11622.
> > 8. Baik S, Hong S, Lee KM. "Learning to Forget for Meta-Learning". 13 Jun 2019. http://arxiv.org/abs/1906.05895
> > 9. Raghu A, Raghu M, Bengio S, Vinyals O. "Rapid Learning or Feature Reuse? Towards Understanding the Effectiveness of MAML". 19 Sep 2019. http://arxiv.org/abs/1909.09157
> > 10. Javed K, Yao H, White M. "Is Fast Adaptation All You Need?". 3 Oct 2019. http://arxiv.org/abs/1910.01705
> > 11. Grefenstette, Edward, Brandon Amos, Denis Yarats, Phu Mon Htut, Artem Molchanov, Franziska Meier, Douwe Kiela, Kyunghyun Cho, and Soumith Chintala. 2019. “Generalized Inner Loop Meta-Learning.” arXiv [cs.LG]. arXiv. http://arxiv.org/abs/1910.01727.
> > 12. Nichol, Alex, Joshua Achiam, and John Schulman. 2018. “On First-Order Meta-Learning Algorithms.” arXiv [cs.LG]. arXiv. http://arxiv.org/abs/1803.02999.
> > 13. Wang, Jane X., Zeb Kurth-Nelson, Dhruva Tirumala, Hubert Soyer, Joel Z. Leibo, Remi Munos, Charles Blundell, Dharshan Kumaran, and Matt Botvinick. 2016. “Learning to Reinforcement Learn.” arXiv [cs.LG]. arXiv. http://arxiv.org/abs/1611.05763.
> > 14. Vanschoren, Joaquin. 2019. “Meta-Learning.” In Automated Machine Learning: Methods, Systems, Challenges, edited by Frank Hutter, Lars Kotthoff, and Joaquin Vanschoren, 35–61. Cham: Springer International Publishing.
> > 15. Rakelly, Kate, Aurick Zhou, Deirdre Quillen, Chelsea Finn, and Sergey Levine. 2019. “Efficient Off-Policy Meta-Reinforcement Learning via Probabilistic Context Variables.” arXiv [cs.LG]. arXiv. http://arxiv.org/abs/1903.08254.
> > 16. Rusu, Andrei A., Dushyant Rao, Jakub Sygnowski, Oriol Vinyals, Razvan Pascanu, Simon Osindero, and Raia Hadsell. 2018. “Meta-Learning with Latent Embedding Optimization.” arXiv [cs.LG]. arXiv. http://arxiv.org/abs/1807.05960.
> > 17. Baydin, Atilim Gunes, Robert Cornish, David Martinez Rubio, Mark Schmidt, and Frank Wood. 2017. “Online Learning Rate Adaptation with Hypergradient Descent.” arXiv [cs.LG]. arXiv. http://arxiv.org/abs/1703.04782.
> > 18. Park, Eunbyung, and Junier B. Oliva. 2019. “Meta-Curvature.” arXiv [cs.LG]. arXiv. http://arxiv.org/abs/1902.03356.

---

### Official Review · AnonReviewer2 · 2019-10-26
**Official Blind Review #2**

**Rating:** 3

**Review:**

This paper analyzes the popular MAML (Model-Agnostic Meta-Learner) method, and thereafter proposes a new approach to meta-learning based on observations from empirical studies. The key idea of the work is to separate the base model and task-specific adaptation components of MAML. This decoupling of adaptation and modeling reduces the burden on the model, thus enabling smaller memory efficient deep learning models to adapt and give high performance on meta learning tasks. The paper proposes a learnable meta-optimizer consisting of a parametrized function U such that the knowledge of adaptation is embedded into its parameters (A,b), instead of forward model parameters. The computational challenges posed by the proposed method are addressed by expressing the parameter matrix A as a Knonecker product of small matrices which is more efficient from memory and time complexity view point. The results on Omniglot and CIFAR-FS are promising, and the paper shows that the proposed meta-optimize is "more expressive", as well as can adapt a shallower model to the same level of performance as MAML.

+ves:
+ The discussion on the deficiency of MAML combined with shallow models is well-supported experimentally.
+ The idea to leverage the parameters of a meta-optimizer for adaptation instead of using model parameters is novel and interesting.
+ The paper is well-written and easy to follow. It motivates its choices well, both in the proposed method and the experiments.
+ The paper presents fair comparison in all experiments with appropriately chosen baseline models, and the proposed approach is validated for both linear as well as non linear models using benchmark datasets.

Concerns:
- While MAML was a seminal work and is widely followed, there have been many follow-ups of MAML, including another widely used method Reptile (Nichol et al, On First-Order Meta-Learning Algorithms). How is the proposed method relevant more broadly to this genre of methods? Some discussion of this would have been useful to understand the generalizability of the idea.

- The choice of the Kronecker product to handle the dimensionality of the meta-optimizer is supported by the paper, but is not very convincing. How important is this choice? What if other decompositions were used?

- The paper seems to state that shallow models are convex (Sec 3.2); however, weight symmetry induces non-convexity even in shallow models. This perspective of the problem may not be very well-justified.

- In Sec 3.2, the paper compares the 1-step adaptation accuracy of a shallow network and a deeper 4 layered linear network and claim that shallow networks underperform. However this underperformance might be due to the difference in required number of steps to reach optimal performance by the two models, and may not be a fair comparison. Why is this conclusive inference? Considering these inferences motivate the full paper, this is important.

- All the presented results are on small CNNs. The paper motivates this as “easing the computational burden”. The original MAML work shows results on state-of-the-art convolutional and recurrent models. It may be important to show results on deeper models to be more confident about its applicability.

- Although one can obtain smaller meta-learned models using the proposed method, training via this method will incur a higher computational burden than MAML-trained deep models. The paper does not talk about this additional complexity at all. Comparisons of wall-clock times or asymptotic analysis of the proposed method w.r.t. MAML would have greatly helped understand the pros and cons of the method.

I am on the borderline on this work - it is a well-written paper with a clear objective and support. But lack of rigorous analysis of the proposed method in terms of the method (how important is the Kronecker factorization?), experiments (with deeper architectures) and a more generalizable understanding of the proposed idea seems to be limiting the work's impact.

========POST-REBUTTAL COMMENTS===============
I thank the authors for their response, and all the efforts in the updated manuscript. Some of the clarifications sought were answered clearly. However, unfortunately, I continue to remain on the borderline on this work for the reasons below. (I would be willing to increase my rating to 4 or 5, which however are not available on the drop down, but perhaps not beyond).

- The response to AnonReviewer1 says that "there have been multiple empirical and theoretical works dedicated solely to the study of MAML [7-11]", hence supporting this work dedicating its focus to MAML alone. However, on close observation, most of these efforts are not published on peer-reviewed avenues and are only on arXiv at this time. Ref [7] (Finn and Levine, 2017) is published but has significantly stronger contributions. Considering the largely empirical nature of this work, showing its generalizability would be required, in my opinion, to make the conclusions of this work useful to the audience. Expecting that it would naturally hold for other methods like REPTILE may not be sufficient. In my opinion, this is a significant limitation.

- I personally remained unconvinced about the response to the question on number of adaptation steps, as well as on the lack of deeper models in the empirical studies.

I once again appreciate the authors for all the additional efforts, it may just be good for the work to be more comprehensive to be relevant and useful.

**Experience Assessment:**

I have published one or two papers in this area.

**Review Assessment: Checking Correctness Of Derivations And Theory:**

I assessed the sensibility of the derivations and theory.

**Review Assessment: Checking Correctness Of Experiments:**

I carefully checked the experiments.

**Review Assessment: Thoroughness In Paper Reading:**

I read the paper at least twice and used my best judgement in assessing the paper.

---

> ### Author Response · Authors · 2019-11-15
> **Response to AnonReviewer2**
>
> We thank AnonReviewer2 for taking the time to write such an extensive review. We will address the reviewer’s concerns point-by-point below.
>
> 1. We believe MAML and many followup works such as Repitle (specifically pointed by you) share a common modeling architecture: the adaptation mechanism shares the same set of networks as the model’s learning weights for encoding inductive bias for the target tasks. Thus, we believe similar dependency on depth would likely be observed (the exact tradeoff would be different).
>
> Note that for drastically different architectures for meta-learning such as RL2 [1], meta-features [2], PEARL [3], the adaptation mechanism is separated (RL2 using the LSTM’s hidden states and PEARL using external embedding space).  Our observation on MAML is not necessarily applicable.
>
> 2. The choice of the Kronecker product over other decomposition methods was mostly motivated by the observation that the identity lies in the span of “Kronecker factorizable” matrices. Concretely, this means that by initializing L, R to the identity, we recover gradient descent as the first adaptation step. In contrast, low-rank factorizations do not span the identity. (By definition, since the identity is full-rank.) This makes it unclear how to initialize the low-rank factors, which can make or break deep learning methods [8] Nonetheless, we report the following results for the Cholesky decomposition in Appendix A.2.3: a rank 1 Cholesky decomposition (SCNN w/ CFC1) gets approximately 70% accuracy on Omniglot, while a rank 10 decomposition (SCNN w/ CFC10) — approximately the same number of parameters as KFC — obtains around 80%. For CIFAR-FS, SCNN w/ CFC1 gets 32% and SCNN w/ CFC10 gets 48%. For mini-ImageNet, SCNN w/ CFC 1 gets 16% and SCNN w/ CFC10 gets 21%.
>
> 3. We indeed state the shallow LR models (i.e. without hidden layers) induce a convex loss in Section 3.2 That is because for a single task, we are trying to solve a logistic regression problem, which is convex. As pointed out in the review, the linear network (LR + LinNet) on the same setting induces a non-convex loss due to overparameterization.
>
> 4. # of adaptation steps for shallow and deep models (section 3.2): both  the shallow and linear network encode a linear decision boundary, they will both obtain comparable performance if properly adapted long enough.  The contrast using the same # of adaptation steps, however,  illustrates the failure mode of MAML: it does not give good initialization points for the shallow model but does give good initialization points for a linear network. In other words, the deeper model is more amenable to adapting, while the two have the same expressiveness.
>
> 5. Regarding the size and type of models in our experiments, we note that we carefully replicated the original classification experiments from the MAML paper. (available at [9]) To the best of our knowledge, these 4-layer CNNs are still widely used and methods that take advantage of larger networks (e.g. ResNet 12, WRN) were specifically designed for such kind of models. (c.f. Table 1 in [10]) Regarding recurrent networks, we are not aware of any work successfully combining them with MAML.
>
> 6. Regarding computational metrics (e.g. time and memory complexity, wall-clock timings), we have added an extensive comparison in Appendix A.4. Asymptotic complexities for the forward pass of the linear optimizer are provided in Section 4.2, and the backward pass has similar complexity as it is computed by back-propagation. Concretely, for a n-layer meta-optimizer, the time complexity of the forward pass grows to O(n*k*sqrt(k)) and the memory complexity to O(nk). As pointed out in the review, our method trades expressivity for computation; when MAML takes 0.63 seconds to compute 1 meta-gradient (on the CIFAR-FS setting) our method takes 2.05 seconds, resulting in a 3.25x slow-down. With more adaptation steps, (e.g. for Omniglot) meta-training with meta-optimizers can be as much as 10x slower than MAML. Note that this slow-down only affects meta-training times and that inference time remains unchanged. For more information, please refer to the table available in Appendix A.4.

---

> > ### Author Response · Authors · 2019-11-15
> > **References**
> >
> > References:
> > 1. Duan, Yan, John Schulman, Xi Chen, Peter L. Bartlett, Ilya Sutskever, and Pieter Abbeel. 2016. “RL2: Fast Reinforcement Learning via Slow Reinforcement Learning.” arXiv [cs.AI]. arXiv. http://arxiv.org/abs/1611.02779.
> > 2. Castiello, Ciro, Giovanna Castellano, and Anna Maria Fanelli. 2005. “Meta-Data: Characterization of Input Features for Meta-Learning.” In Modeling Decisions for Artificial Intelligence, 457–68. Springer Berlin Heidelberg.
> > 3. Rakelly, Kate, Aurick Zhou, Deirdre Quillen, Chelsea Finn, and Sergey Levine. 2019. “Efficient Off-Policy Meta-Reinforcement Learning via Probabilistic Context Variables.” arXiv [cs.LG]. arXiv. http://arxiv.org/abs/1903.08254.
> > 4. Baydin, Atilim Gunes, Robert Cornish, David Martinez Rubio, Mark Schmidt, and Frank Wood. 2017. “Online Learning Rate Adaptation with Hypergradient Descent.” arXiv [cs.LG]. arXiv. http://arxiv.org/abs/1703.04782.
> > 5. Nichol, Alex, Joshua Achiam, and John Schulman. 2018. “On First-Order Meta-Learning Algorithms.” arXiv [cs.LG]. arXiv. http://arxiv.org/abs/1803.02999.
> > 6. Rothfuss, Jonas, Dennis Lee, Ignasi Clavera, Tamim Asfour, and Pieter Abbeel. 2018. “ProMP: Proximal Meta-Policy Search.” http://arxiv.org/abs/1810.06784.
> > 7. https://github.com/openai/supervised-reptile/#reproducing-training-runs
> > 8. Glorot, X., and Y. Bengio. 2010. “Understanding the Difficulty of Training Deep Feedforward Neural Networks.” Proceedings of the Thirteenth International Conference. http://www.jmlr.org/proceedings/papers/v9/glorot10a/glorot10a.pdf?hc_location=ufi.
> > 9. https://github.com/cbfinn/maml
> > 10. Lee, Kwonjoon, Subhransu Maji, Avinash Ravichandran, and Stefano Soatto. 2019. “Meta-Learning with Differentiable Convex Optimization.” arXiv [cs.CV]. arXiv. http://arxiv.org/abs/1904.03758.

---

### Official Review · AnonReviewer3 · 2019-11-01
**Official Blind Review #3**

**Rating:** 6

**Review:**

This paper presents an experimental study of gradient based meta learning models and most notably MAML. The results suggest that modeling and adaptation are happening on different parts of the network leading to an inefficient use of the model capacity which explains the poor performance of MAML on linear (or small networks) models. To tackle this issue they proposed a kronecker factorization of the meta optimizer.

The paper is well motivated and well written in terms of clarity in the message and being easy to follow.

One major issue is that the experimental study is not that comprehensive to support the claim of the paper. Especially, in analyzing the failure case of linear models.For example, one may try small (but nonlinear networks) and compare its performance with larger (possibly overparameterized) ones on at least 2 standard network architectures. But, it doesn't mean that I don't like the paper at its current state. The paper yet has a message and it's delivered clearly.

I wonder if the overparameterized is just related to depth or overparameterization in width would work too? If not then it might be the "nonlinearity" that is doing the work

In section 3.2 (Figure 2, left) and (Figure2, mid) show that FC follows the pattern of C1-C3. t
Then the authors proposed the experiment related to perturbing FC (Figure 2, right) to show that FC is actually not similar to C1-C3 and is important to adaptation. However, one can do similar experiments for C1-C3 and claim they are also important to adaptation. It seems that FC and C4 are really different.

For a non-expert reader it's not readily clear that how the kronecker factorization of A leads to equation 5. An explanation can help. Also, a few sentences or schematic demonstration of kronecker product makes the paper self-contained.

There are a few typos in the paper that can be removed after a thorough proofreading.

**Experience Assessment:**

I have read many papers in this area.

**Review Assessment: Checking Correctness Of Derivations And Theory:**

I assessed the sensibility of the derivations and theory.

**Review Assessment: Checking Correctness Of Experiments:**

I assessed the sensibility of the experiments.

**Review Assessment: Thoroughness In Paper Reading:**

I read the paper thoroughly.

---

> ### Author Response · Authors · 2019-11-15
> **Response to AnonReviewer3**
>
> We thank AnonReviewer3 for their review.
>
> We have substantially added to the Appendix in order to clarify some of our results.
>
> Regarding the issue of overparameterization of depth vs width, we have added extensive results in Appendix A.2.1 where we trained the binary linear network with varying width (w=2, 4 … , 256) and depth (l=1, 2, 3, 4). We observe that the linear network is always able to adapt and solve the tasks regardless of the width of the hidden layers, so long as the model has at least one hidden layer.
>
> A discussion of the difference in behaviour between C1-C3 and FC is provided in Appendix A.2.2. For each layer of a meta-trained model, we scale the weights of the layer by a given factor before fast-adaptation. We observe that for C1-C3, this scaling does not impact the post-adaptation accuracy. However, for C4 and FC, scaling weights pre-adaptation is catastrophic for post-adaptation accuracy: by perturbing those layers, the model is not able to compute a fast-adapting update and its post-adaptation accuracy drops to chance. For more details, including a discussion of post-adaptation scaling, please refer to Appendix A.2.2.
>
> On the effect of non-linearity enabling fast-adaptation, we point out that all models in Section 5.2 use non-linearities. Yet, while they are able to adapt better than chance, the non-linearity does not allow them to perform as well as deeper models.
>
> As for the expository issues, we have added references [1, Section 9.1; 2, Section 10.2.2] to the derivation of Equation 5, a schematic of the Kronecker product, and a schematic and pseudo-code for our proposed method. Those are available in Appendix A.3.

---

> > ### Author Response · Authors · 2019-11-15
> > **References**
> >
> > References:
> > 1. Bernstein, Dennis S. 2018. Scalar, Vector, and Matrix Mathematics: Theory, Facts, and Formulas - Revised and Expanded Edition. Revised, Expanded edition. Princeton University Press.
> > 2. Petersen, Kaare Brandt, and Michael Syskind Pedersen. n.d. “The Matrix Cookbook.” Perrylea.com. http://www.perrylea.com/Perry_and_Dawns_Home_Page/Free_Engineering_and_Math_Text_files/Matrix%20Cookbook.pdf.

---

### Public Comment · ~Mikhail_Khodak1 · 2019-10-25
**Understanding the negative results for convex linear models**

This submission makes the interesting claim that initialization-based meta-learning algorithms require over-parameterized models to learn a good starting point. Given that a significant part of the motivation for this work is made using an empirical analysis of the linear case, I think that a set of recent efforts studying exactly this setting [1,2,3] is quite relevant (note: I am a co-author on one of these papers), especially because most of the theoretical results are positive, whereas the experimental results for the (convex) linear case presented in this submission are negative. Note that the discussion below is meant not as a challenge to the motivational claim, which seems plausible, but as a theorist's effort to understand whether our assumptions are failing/whether studying the over-parameterized case can yield better bounds.

In particular, Finn et al. [1] make an argument in support of the (convex) linear setting [1, Appendix A] and show learnability of the MAML base-learner [1, Corollaries 1 & 2], specifically that optimizing the MAML objective yields an initialization whose error converges to that of the optimal initialization as you see more tasks. This goes against the results in Figure 1, especially the left plot: it is possible that the over-parameterization leads to a better optimization geometry and thus better post-adaptation results, but the inability to exceed random accuracy at all is surprising to me. It would help to have answers to the following questions:
    1. How many shots were used for the experiments on synthetic data, and how does the relative performance improve with more shots? Finn et al. [1] assume strong-convexity, which will fail in the few-shot setting (number of samples < input dimension), but it is unclear if this assumption is necessary.
    2. In A.1.1 it says "Due to the argument presented in the main text, any hyper-parameter setup will replicate the logistic regression (LR) results, but we used meta and adaptation learning rates of 0.01 and 0.5." Where/what is this argument, and does this mean learning rates were not tuned? Existing theory depends strongly on these learning rates [1,2,3].

The analyses in the other papers [2,3] deal with (variants of) Reptile, which while less well-known is still quite popular; it would be interesting/surprising if the results in the submission were true for MAML but not Reptile. Furthermore, the Reptile results do not assume strong-convexity. They do all depend on tasks-similarity, i.e. that linear classifiers that perform well on different tasks are close together; the submission argues in Section 3.2 that this may not be true in practice. On the other hand, both Denevi et al. [2] and Khodak et al. [3] report positive experimental results with (convex) linear models. Denevi et al. [2] also include an evaluation on synthetic data, which may be a useful comparison, while Khodak et al. [3, Figure 1] show that the optimal linear classifiers on a toy text classification task are indeed close together. So perhaps the claimed need for over-parameterization depends strongly on properties of the data that do not always hold in settings where we might want to use linear models.


Minor Point:
I suggest a rephrasing of the following statement from Section 3.2, as convex functions may have zero, one, or infinitely many optima: "however, if the model is shallow such that L_τ is convex in its parameters, then any initialization that is good for fast adapting to one subset of tasks could be bad for another subset of tasks since all the tasks have precisely one global minimizer and those minimizers can be arbitrarily far from each other."


References:
[1] Finn, Rajeswaran, Kakade, Levine. Online Meta-Learning. ICML 2019.
[2] Denevi, Ciliberto, Grazzi, Pontil. Learning-to-Learn Stochastic Gradient Descent with Biased Regularization. ICML 2019.
[3] Khodak, Balcan, Talwalkar. Provable Guarantees for Gradient-Based Meta-Learning. ICML 2019.

---

> ### Author Response · Authors · 2019-11-15
> **Response to Khodak**
>
> Thank you for your interest, we look forward to complementary theoretical explanations to the questions this manuscript raises.
>
> Our introduction omits a theoretical discussion of the synthetic binary classification experiment for expository reasons: as explained below, our experimental setup differs from the existing literature making comparisons difficult.
>
> The remaining of this answer addresses the questions point-by-point.
>
> Q1: How do you reconcile the experimental results results in Figure 1 with the theory in Finn et al. [Corollaries 1 & 2] ?
>
> To the best of our knowledge, existing theoretical literature for MAML requires strong convexity of the MAML loss. This is the case for Finn et al. (Assumption 2) as well as Khodak et al. and Denevi et al. in the form of L2 regularization. In our binary classification experiments, we directly minimize the binary cross-entropy -- which is not strongly convex -- and violate the assumptions in those works. For example, to match the assumptions in Finn et al., we would have to set the fast adaptation learning rate to 0 thus recovering the multi-task scenario.
>
> While we can not comment on the failure of the assumptions, we can provide the following insights. As pointed out in the manuscript, when initializing the weights of the convex model at the origin we obtain a high (>90%) post-adaptation accuracy. However, and despite our best hyper-parameter tuning efforts, reaching that point seems infeasible via gradient descent; in other words, shallow models can be hard to meta-learn. This issue is delicate to diagnose, as the training difficulty is induced by the MAML loss (non-convexity) and its evaluation (stochasticity). We note that when including L2-regularization in the MAML loss, meta-learning of the convex model becomes possible and the model reaches approximately 90% accuracy.
>
> Q2: How many shots were used, and how does performance improve with more shots ?
>
> At every timestep we sample a new dataset consisting of 1,000 data points in $\mathbb{R}^{100}$, and allow for 1 adaptation step. The meta-batch size is set to 1. In preliminary experiments, using 10x more data points does not improve learnability.
>
> Q3: What is the argument mentioned in the Appendix ? Does that mean the learning rate were not tuned for that experiment ?
>
> This sentence in the Appendix is indeed poorly phrased and we have modified it. Naturally, all learning rates in this experiment were tuned to the best of our ability. The argument we refer to is the (empirical) one presented in Section 3.2.
>
> Q4: Do the experimental results in Figure 1 also hold for Reptile, or only MAML ?
>
> We do not have results for Reptile, but we believe that our conclusions apply. (c.f. response to AnonReviewer2.)
>
> References:
> 1. Finn, Rajeswaran, Kakade, Levine. Online Meta-Learning. ICML 2019.
> 2. Khodak, Balcan, Talwalkar. Provable Guarantees for Gradient-Based Meta-Learning. ICML
> 3. Denevi, Ciliberto, Grazzi, Pontil. Learning-to-Learn Stochastic Gradient Descent with Biased Regularization. ICML 2019.

---

### Decision · Program_Chairs · 2019-12-19

**Decision:**

Reject

**Comment:**

This paper presents a number of experiments involving the Model-Agnostic Meta-Learning (MAML) framework, both for the purpose of understanding its behavior and motivating specific enhancements.  With respect to the former, the paper argues that deeper networks allow earlier layers to learn generic modeling features that can be adapted via later layers in a task-specific way.  The paper then suggests that this implicit decomposition can be explicitly formulated via the use of meta-optimizers for handling adaptations, allowing for simpler networks that may not require generic modeling-specific layers.

At the end of the rebuttal and discussion phases, two reviewers chose rejection while one preferred acceptance.  In this regard, as AC I did not find clear evidence that warranted overriding the reviewer majority, and consistent with some of the evaluations, I believe that there are several points whereby this paper could be improved.

More specifically, my feeling is that some of the conclusions of this paper would either already be expected by members of the community, or else would require further empirical support to draw more firm conclusions.  For example, the fact that earlier layers encode more generic features that are not adapted for each task is not at all surprising (such low-level features are natural to be shared).  Moreover, when the linear model from Section 3.2 is replaced by a deep linear network, clearly the model capacity is not changed, but the effective number of parameters which determine the gradient update will be significantly expanded in a seemingly non-trivial way.  This is then likely to be of some benefit.

Consequently, one could naturally view the extra parameters as forming an implicit meta-optimizer, and it is not so remarkable that other trainable meta-optimizers might work well.  Indeed cited references such as (Park & Oliva, 2019) have already applied explicit meta-optimizers to MAML and few-shot learning tasks.  And based on Table 2, the proposed factorized meta-optimizer does not appear to show any clear advantage over the meta-curvature method from (Park & Oliva, 2019).  Overall, either by using deeper networks or an explicit trainable meta-optimizer, there are going to be more adaptable parameters to exploit and so the expectation is that there will be room for improvement.  Even so, I am not against the message of this paper.  Rather it is just that for an empirically-based submission with close ties to existing work, the bar is generally a bit higher in terms of the quality and scope of the experiments.

As a final (lesser) point, the paper argues that meta-optimizers allow for the decomposition of modeling and adaptation as mentioned above; however, I did not see exactly where this claim was precisely corroborated empirically.  For example, one useful test could be to recreate Figure 2 but with the meta-optimizer in place and a shallower network architecture. The expectation then might be that general features are no longer necessary.